# Integrating contact tracing and whole-genome sequencing to track the elimination of dog-mediated rabies: An observational and genomic study

Kennedy Lushasi[1,2,3†], Kirstyn Brunker[2†], Malavika Rajeev[4], Elaine A Ferguson[2], Gurdeep Jaswant[1,5], Laurie Louise Baker[2,6], Roman Biek[2], Joel Changalucha[1,2,7], Sarah Cleaveland[2], Anna Czupryna[2], Anthony R Fooks[7], Nicodemus J Govella[1], Daniel T Haydon[2], Paul CD Johnson[2], Rudovick Kazwala[8], Tiziana Lembo[2], Denise Marston[7,9], Msanif Masoud[10], Matthew Maziku[11], Eberhard Mbunda[11], Geofrey Mchau[12], Ally Z Mohamed[13], Emmanuel Mpolya[3], Chanasa Ngeleja[14], Kija Ng'habi[15], Hezron Nonga[11], Kassim Omar[13], Kristyna Rysava[16], Maganga Sambo[1], Lwitiko Sikana[1], Rachel Steenson[2], Katie Hampson[2]*

[1]Environmental Health and Ecological Sciences Department, Ifakara Health Institute, Dar es salaam, United Republic of Tanzania; [2]Boyd Orr Centre for Population and Ecosystem Health, School of Biodiversity, One Health & Veterinary Medicine, University of Glasgow, Glasgow, United Kingdom; [3]Department of Global Health and Biomedical Sciences, School of Life Sciences and Bioengineering, Nelson Mandela African Institution of Science and Technology, Arusha, United Republic of Tanzania; [4]Department of Ecology and Evolutionary Biology, Princeton University, Princeton, United States; [5]The University of Nairobi, Nairobi, Kenya; [6]College of the Atlantic, Bar Harbor, United States; [7]Animal & Plant Health Agency, Weybridge, United Kingdom; [8]Department of Veterinary Medicine and Public Health, Sokoine University of Agriculture, Morogoro, United Republic of Tanzania; [9]Department of Comparative Biomedical Sciences, School of Veterinary Medicine, University of Surrey, Guildford, Surrey, United Kingdom; [10]Ministry for Health and Social Welfare, Zanzibar, United Republic of Tanzania; [11]Ministry of Livestock Development and Fisheries, Dodoma, United Republic of Tanzania; [12]Department of Epidemiology, Ministry of Health, Community Development, Gender, Elderly and Children (MoHCDGEC), Dodoma, United Republic of Tanzania; [13]Department of Livestock Development, Pemba, Ministry of Livestock Development and Fisheries, Zanzibar, United Republic of Tanzania; [14]Tanzania Veterinary Laboratory Agency, Dar es salaam, United Republic of Tanzania; [15]Mbeya college of Health and Allied Sciences, University of Dar es Salaam, Mbeya, United Republic of Tanzania; [16]University of Warwick, Coventry, United Kingdom

**\*For correspondence:**
katie.hampson@glasgow.ac.uk

[†]These authors contributed equally to this work

**Competing interest:** The authors declare that no competing interests exist.

## Abstract

**Background:** Dog-mediated rabies is endemic across Africa causing thousands of human deaths annually. A One Health approach to rabies is advocated, comprising emergency post-exposure vaccination of bite victims and mass dog vaccination to break the transmission cycle. However, the impacts and cost-effectiveness of these components are difficult to disentangle.

**Methods:** We combined contact tracing with whole-genome sequencing to track rabies transmission in the animal reservoir and spillover risk to humans from 2010 to 2020, investigating how the components of a One Health approach reduced the disease burden and eliminated rabies from Pemba Island, Tanzania. With the resulting high-resolution spatiotemporal and genomic data, we inferred transmission chains and estimated case detection. Using a decision tree model, we quantified the public health burden and evaluated the impact and cost-effectiveness of interventions over a 10-year time horizon.

**Results:** We resolved five transmission chains co-circulating on Pemba from 2010 that were all eliminated by May 2014. During this period, rabid dogs, human rabies exposures and deaths all progressively declined following initiation and improved implementation of annual islandwide dog vaccination. We identified two introductions to Pemba in late 2016 that seeded re-emergence after dog vaccination had lapsed. The ensuing outbreak was eliminated in October 2018 through reinstated islandwide dog vaccination. While post-exposure vaccines were projected to be highly cost-effective ($256 per death averted), only dog vaccination interrupts transmission. A combined One Health approach of routine annual dog vaccination together with free post-exposure vaccines for bite victims, rapidly eliminates rabies, is highly cost-effective ($1657 per death averted) and by maintaining rabies freedom prevents over 30 families from suffering traumatic rabid dog bites annually on Pemba island.

**Conclusions:** A One Health approach underpinned by dog vaccination is an efficient, cost-effective, equitable, and feasible approach to rabies elimination, but needs scaling up across connected populations to sustain the benefits of elimination, as seen on Pemba, and for similar progress to be achieved elsewhere.

**Funding:** Wellcome [207569/Z/17/Z, 095787/Z/11/Z, 103270/Z/13/Z], the UBS Optimus Foundation, the Department of Health and Human Services of the National Institutes of Health [R01AI141712] and the DELTAS Africa Initiative [Afrique One-ASPIRE/DEL-15-008] comprising a donor consortium of the African Academy of Sciences (AAS), Alliance for Accelerating Excellence in Science in Africa (AESA), the New Partnership for Africa's Development Planning and Coordinating (NEPAD) Agency, Wellcome [107753/A/15/Z], Royal Society of Tropical Medicine and Hygiene Small Grant 2017 [GR000892] and the UK government. The rabies elimination demonstration project from 2010-2015 was supported by the Bill & Melinda Gates Foundation [OPP49679]. Whole-genome sequencing was partially supported from APHA by funding from the UK Department for Environment, Food and Rural Affairs (Defra), Scottish government and Welsh government under projects SEV3500 and SE0421.

## Editor's evaluation

In this work, the authors set out to use contact tracing and whole-genome sequencing to track the elimination of dog-mediated rabies in Pemba Island, Tanzania. A major strength is the use of multiple data types in the analysis. The work will likely have an impact on influencing the practical policies that can be implemented to target the elimination of dog-mediated rabies in other regions/contexts.

## Introduction

Every year an estimated 59,000 people die from rabies (*Hampson et al., 2015*), a viral infection transmitted primarily by domestic dogs in low- and middle-income countries (LMICs). While human rabies encephalitis remains incurable, the disease is readily preventable if post-exposure prophylaxis (PEP) is promptly administered to bite victims upon exposure (*World Health Organization, 2018b*). Moreover, mass dog vaccination has eliminated dog-mediated rabies from high-income countries and much of the Americas (*Vigilato et al., 2013*). Yet, in most African and Asian countries there has been little investment in dog vaccination and rabies circulates unabated (*Hampson et al., 2015*). A global goal to eliminate human deaths from dog-mediated rabies by 2030 ('Zero by 30') is now advocated (*Minghui et al., 2018*), with recommendations to scale up mass dog vaccination.

Although dog vaccination can eliminate dog-mediated rabies, there are challenges to achieving this goal. In most rabies endemic countries in sub-Saharan Africa, dog vaccination campaigns have

been sparse and localised (*World Health Organization, 2018a*). Moreover, the high reproductive rates and short lifespan of dogs in many LMICs quickly lead to drops in vaccination coverage, with repeat campaigns required to maintain coverage (*Davlin and Vonville, 2012*). The virus can easily spread in dog populations that have low and heterogeneous vaccination coverage (*Mancy et al., 2022*) and incursions leading to outbreaks are commonly reported (*Bourhy et al., 2016*; *Zinsstag et al., 2017*; *Rysava et al., 2020*), often facilitated by human-mediated movement of dogs incubating infection (*Townsend et al., 2013b*; *Tohma et al., 2016*). This situation is compounded by weak surveillance which hinders effective outbreak response and poses a challenge for monitoring progress towards elimination, including how to determine disease freedom (*Nel, 2013*).

Across the African continent there are very few documented examples of elimination of dog-mediated rabies. We found just four papers reporting locations on the continent with potential interruption of transmission by dog vaccination; the cities of N'Djamena, Chad (*Zinsstag et al., 2017*) and Harare, Zimbabwe (*Coetzer et al., 2019*), Serengeti district in northwest Tanzania (*Cleaveland et al., 2003*) and KwaZulu-Natal province in South Africa (*Sabeta and Ngoepe, 2018*). In all four locations, endemic circulation has since re-established, with resurgences explained by movement of infected dogs from surrounding areas after dog vaccination campaigns lapsed. The importance of reintroductions in maintaining rabies circulation is further highlighted from long-term surveillance from Bangui, the capital of the Central African Republic (*Bourhy et al., 2016*) and from long-term contact tracing in Serengeti district, Tanzania (*Mancy et al., 2022*). Genomic surveillance can potentially play a role in differentiating rabies introductions from undetected sustained transmission, and thus in confirming or refuting rabies elimination and therefore targeting of control efforts. However, sequencing of rabies viruses also remains limited in Africa.

Dog-mediated rabies is endemic in East Africa where thousands of human rabies deaths occur each year (*Hampson et al., 2019*). Rabies has circulated on Pemba Island, off mainland Tanzania, since the late 1990s. Dog vaccinations on Pemba first began in 2010, with a small-scale campaign conducted by the animal welfare organisation, World Animal Protection (formerly WSPA). Over the next 5 years, a rabies elimination demonstration project, funded by the Bill & Melinda Gates Foundation, coordinated by the World Health Organisation and led by the Tanzanian government, was implemented across southeast Tanzania, including Pemba (*Mpolya et al., 2017*). Here, we show how these efforts led to rabies elimination, while highlighting how introductions pose challenges to achieving and maintaining rabies-freedom even on a small, relatively isolated, island. Our study is the first to confirm rabies elimination from an African setting, including in response to reintroduction, through quantifying case detection. Using rigorous contact tracing, we identified chains of transmission within the domestic dog reservoir informed by in-country whole-genome sequencing (the first example in Africa) and cross-species transmission from domestic dogs to humans. This enabled us to estimate the public health burden and associated cost-effectiveness of both post-exposure vaccination and dog vaccination, as well as their combined use, in achieving and maintaining rabies freedom on Pemba. Our findings illustrate the critical need to holistically link surveillance with public health and veterinary interventions to cost-effectively reduce the burden of zoonotic pathogens. This case study provides timely lessons given the global strategic plan to eliminate dog-mediated human rabies by 2030.

## Methods

### Study population

Pemba (988 km$^2$) is situated fifty kilometres from the Tanzanian mainland. The island comprises four administrative districts with 121 villages (*shehias*) and a projected human population of 438,765 in 2020 (*National Bureau of Statistics Tanzania, 2012*). The human: dog ratio is very high (~118 humans to 1 dog), in this predominantly muslim population. Almost all dogs on Pemba are unconfined and 10–20% are thought to be unowned, potentially posing a problem for reaching the vaccination coverage needed for elimination using central point vaccination strategies.

### Epidemiological and laboratory investigations

Records of bite patients presenting to health facilities and of suspect or probable rabid animals reported to the district veterinary offices on Pemba Island from January 2010 until January 2021 were used to initiate contact tracing (*Mancy et al., 2022*). Bite victims and, if known, the owners of

biting animals were exhaustively traced, recording details of all bite incidents, including dates and coordinates. Other people or animals that were identified as bitten were further traced. The status of animals was assessed from their reported behaviour and outcome (whether they died, disappeared or survived), and classified according to WHO case definitions (*World Health Organization, 2018a*). Briefly, an animal showing any clinical signs of rabies was considered a suspect case; if a suspect case had a reliable history of contact with a suspect rabid animal and/or was killed, died or disappeared within 10 days of observation of illness, the animal was considered a probable case. Animals that remained alive for more than 10 days after biting a person, were considered healthy. Brain tissue samples were collected from animal carcasses for diagnostic testing whenever possible (*Rupprecht et al., 2018*).

Two batches of sequencing were performed to obtain 16 near whole-genome sequences (WGS) of rabies virus (RABV) from dog brain samples collected on Pemba, with the approach changing as protocols and capacity for in-country sequencing developed (*Brunker et al., 2020*). Eight of these sequences have been previously published within a methods paper (*Brunker et al., 2020*) and 8 are published for the first time here. The latter are archived 2011/12 samples (3) and samples (5) from early outbreak surveillance (September/October 2016) that were confirmed RABV positive at Pemba Veterinary Laboratory Department and shipped to the Animal & Plant Health Agency (APHA), UK. Total RNA was extracted using Trizol (Invitrogen) and a real-time PCR assay (*Marston et al., 2019*) was performed to confirm the presence of RABV and indicate viral load. Metagenomic sequencing libraries were prepared and sequenced on an Illumina MiSeq as previously described (*Brunker et al., 2015*). Subsequent sequencing of the 8 additional samples (September 2016 to May 2017) was conducted in-country in August 2017 at the Tanzania Veterinary Laboratory Agency (TVLA) following an end-to-end protocol using a multiplex PCR approach (*Quick et al., 2016*) for MinION (Oxford Nanopore Technology, Oxford, UK) sequencing of RABV genomes (*Brunker et al., 2020*). Fourteen previously unpublished WGS (via the metagenomic approach) from mainland Tanzania (2009 to 2017) are also published here and included in analyses. The newly published sequences are detailed in *Supplementary file 1*.

## Control and prevention measures

We compiled data on rabies control and prevention measures implemented on Pemba, including numbers and timing of dog vaccination campaigns, and costs of dog vaccination and PEP provisioning (*Supplementary file 2*).

Briefly, the first small-scale dog vaccination campaign (705 dogs vaccinated) on Pemba took place in 2010. This was followed by four annual islandwide campaigns from 2011 through to 2014 carried out by livestock field officers under Pemba's department of livestock as part of the elimination demonstration project (*Mpolya et al., 2017*). One week before each campaign, a meeting was held between District Veterinary Officers, Livestock Field Officers (LFOs), and Community Animal Health Workers (CAHWs) to review protocols and distribute vaccination equipment. CAHWs for each *shehia* then moved door-to-door inviting owners to bring their dogs to the nearest vaccination point and distributed posters. One day before the campaign, CAHWs walked repeatedly through each *shehia* announcing the forthcoming vaccination over a loudspeaker. Vaccination points were mostly situated in the centre of *shehias* but for small neighbouring *shehias*, vaccination points were located at central convenient locations. Each point was operated by two LFOs and a CAHW and campaigns ran from 9.00am to 3.00pm on a single day with vaccinations provided free-of-charge. During the 2013 and 2014 campaigns, dogs were marked with temporary collars upon vaccination and post-vaccination transects were carried out in each *shehia* to estimate coverage achieved.

As part of the demonstration project PEP was procured for free provisioning at Pemba's four district hospitals. Training in administering both intradermal and intramuscular post-exposure vaccination was completed in early 2011. Following the end of the demonstration project in 2015, bite patients were required to pay 30,000 TSh ($12.9) per vial when undergoing post-exposure vaccination, with multiple vials required for a complete PEP course.

In late 2016, a rabies outbreak was detected. The initial government response involved conducting central point dog vaccination campaigns in *shehias* reporting cases. However, these efforts were limited. Island-wide vaccination campaigns were therefore conducted from 2017 onwards, including door-to-door vaccination in some *shehias* where dog owners could not bring dogs to allocated central

points. In 2017, the government of Zanzibar also began to subsidise PEP, making vaccines free-of-charge at Pemba's main hospital and in hospitals in Zanzibar (1 day's ferry travel), otherwise, post-exposure vaccines were available to purchase on the mainland.

## Analyses
### Dog population and vaccination coverage

To estimate time-varying vaccination coverage at the *shehia* level, it was necessary to first estimate dog population sizes. This was achieved using two datasets: (1) government dog population surveys for the years 2012 and 2017–2019, and (2) post-vaccination transects from the 2013 to 2014 vaccination campaigns together with associated numbers of dogs vaccinated in the preceding campaigns. Where at least one collared (i.e. vaccinated) dog and >10 total dogs were observed on a transect, the dog population of a *shehia* at the time of the transect was estimated as:

$$D = \frac{V_d(1+PAR)}{\left(\frac{C_d}{(C_d+U_d)}\right)}$$

where $D$ is the dog population size, $V_d$ is the number of dogs vaccinated in the campaign preceding the transect, $C_d$ is collared dogs, $U_d$ is unmarked dogs, and $PAR$ is the ratio of pups (<3 months) to adult dogs (*Sambo et al., 2018*). $PAR$ was estimated to be 0.256 from a census of the Serengeti District dog population in Northern Tanzania between 2008–2016 (*Sambo et al., 2017*). By multiplying by (1+PAR), we assume both that vaccination campaigns fail to reach pups, and that pups are not counted during transects (*Sambo et al., 2018*).

At least one Government or transect-based dog population estimate was available for each *shehia*, with some having estimates at up to six time points. For each *shehia*, the dog population in every month throughout the study period for which we did not already have an estimate was then projected. For months that lay between two known population estimates, a population projection was obtained via the exponential growth rate calculated between those two estimates. For months where there was only a preceding or subsequent dog population estimate available, we projected the population based on a human:dog ratio calculated from this preceding/subsequent estimate and the human population projected from the 2012 national census (*National Bureau of Statistics Tanzania, 2012*). In some cases, the projected dog population obtained for a month using this approach was lower than the number of dogs vaccinated during a campaign in that month. Where this occurred, the population estimates were adjusted as necessary to prevent coverage estimates exceeding 100%.

The coverage achieved by each vaccination campaign in each *shehia* was obtained by dividing the number of dogs vaccinated by the estimated dog population for the month when the campaign occurred. We estimated the monthly number of dogs with vaccine-induced immunity as follows:

$$\lambda = e^{-\left(\frac{1}{v}+d\right)\left(\frac{1}{12}\right)} \min\left(1, \frac{D_m}{D_{m-1}}\right)$$

$$P_m = \begin{cases} \max(0, V_{m-1}\lambda - N_m), & \textit{if January} \\ \max(0, P_{m-1}\lambda - N_m), & \textit{if any other month} \end{cases}$$

$$V_m = \min(D_m, V_{m-1}\lambda + \max(0, N_m - P_{m-1}\lambda))$$

where $V_m$ is the number of immune dogs at month $m$, $N_m$ is the number of newly vaccinated dogs at $m$, $D_m$ is the dog population at $m$ estimated using the methods described above, and $P_m$ is the number of immune dogs that were vaccinated during campaigns in previous years, not in the current year. Immunity wanes according to both $v$, the mean duration of vaccine-induced immunity (assumed to be 3 years), and $d$=0.595, the annual dog death rate (*Czupryna et al., 2016*). This approach conservatively assumes both that dogs that are immune from previous campaigns are preferentially vaccinated in subsequent campaigns and that, if the dog population declines between months, then this is a consequence of an above average death rate, rather than a below average birth rate. It also assumes that any top-up campaigns in a *shehia* in the current year focus on vaccinating susceptible dogs, avoiding re-vaccination of already vaccinated animals.

## Phylogenetics

Sequence data were used to understand the source and timing of introductions to Pemba and to resolve transmission chains. Raw sequence reads were processed and underwent quality control filtering (*Brunker et al., 2020*; *Brunker et al., 2015*). Pemba sequences were submitted to RABV-GLUE to determine which global RABV subclade they belonged to (*Campbell et al., 2022*). Clade assignment indicated that all Pemba sequences grouped within the RABV minor clade Cosmopolitan-AF1b. Therefore, an exploratory dataset of publically available genome sequences (coverage >90% of genome) from the Cosmopolitan-AF1b clade was obtained from RABV-GLUE (n=244) and supplemented with new sequences published in this paper (n=22, *Supplementary file 1*). Since the genome region and number of sequences varied widely in publically available data, an additional analysis was undertaken using an alignment, downloaded from RABV-GLUE, of all Cosmopolitan-AF1b sequences up to the year 2017 (inclusive) regardless of genome position or length.

For the whole genome sequences, an alignment was created in MAFFT (*Nakamura et al., 2018*) and used to build a maximum likelihood (ML) phylogeny in IQ-TREE (*Nguyen et al., 2015*) with default model selection. To simplify and focus analysis on Pemba outbreak cases, a subtree encompassing all 2016/17 Pemba sequences and relevant contextual sequences was extracted from the ML phylogeny and these sequences were used for Bayesian phylogeographic analysis in BEAST (*Suchard et al., 2018*) For the BEAST analysis, one sequence from Uganda and one sequence from Rwanda were removed from the subset to avoid influencing phylogeographic analysis as the only two non-Tanzania sequences. Two sequences (GenBank accessions: MN726823, MN726822) were also removed as they contained a high proportion of masked bases (Ns) that affected tree convergence. This resulted in a reduced dataset of 153 sequences, exclusively from Tanzania, spanning the years 2001–2017. Note that this excluded two historical, previously published Pemba sequences (2010/12) belonging to a divergent lineage, previously defined as Tz5 (*Brunker et al., 2015*). TempEst was used to assess the temporal signal in the data, with a moderate association between genetic distances and sampling dates ($R^2$=0.37) indicating suitability for phylogenetic molecular clock analysis in BEAST (*Rambaut et al., 2016*).

A Bayesian discrete phylogeographic analysis was conducted in BEAST v1.10.4 on the 153 Tanzanian RABV genomes, of which 13 were from the 2016/17 Pemba outbreak. Two independent MCMC chains were run for 250 million steps with an uncorrelated log-normal relaxed molecular clock. Sequences were partitioned into concatenated coding sequence and non-coding sequence, each with a GTR +G substitution model. Two locations were specified for phylogeographic analysis, 'Mainland' or 'Island' for identifying the source of introductions. Sampled trees were subset to 10,000 trees and summarised as a maximum clade credibility tree, which was examined to determine the timing of introductions. Phylogenies were visualised and annotated in R using the ggtree package (*Yu, 2020*).

The full dataset (i.e. all available sequences) extracted from RABV-GLUE was combined with the new sequences generated in this paper (n=22, *Supplementary file 1*) using MAFFT's function to add new sequences to an existing alignment. R was used to categorise data into sequence types as follows: partial gene length sequences typically obtained from polymerase chain reaction (PCR) based diagnostic assays, full length (>90% coverage) gene sequences (gene) and whole (>90% coverage) genome sequences (WGS). Sequences were further categorised into genome position by gene: nucleoprotein (n), phosphoprotein (p), matrix protein (m), glycoprotein (g), RNA polymerase (l). This facilitated detailed exploration of publically available RABV sequence data to obtain the most informative datasets to compare Pemba outbreak sequences. Background ML phylogenies were produced in IQ-TREE with default settings, using alignments of the variable length gene sequences. Extreme outliers with long branches (upper and lower 1st percentile of branch length distribution) were removed and a subtree extracted stemming from the most recent common ancestor (or one node back from) of all Pemba (historical and outbreak) sequences. Sequences from these subtrees (N and G gene) were subject to a more robust phylogenetic reconstruction with rapid bootstrapping in IQ-TREE and an outgroup sequence from the Cosmopolitan-AF1a minor clade (GenBank Accession: KC196743).

## Transmission trees

Using the case data, we reconstructed transmission trees building on previously described methods (*Mancy et al., 2022*). Traced progenitors were assigned, otherwise links between cases were inferred

probabilistically from dispersal kernel and serial interval distributions incorporating uncertainties in timings. We used distributions previously parameterized from contact tracing in northwest Tanzania (Lognormal serial interval, meanlog 2.85, sdlog 0.966, n=1107 rabid dog case histories; Weibull distance kernel, shape 0.698, scale 1263.954, n=6626 rabid dog biting incidents, with 3275 right-censored due to the unknown start location of the biting dog) (*Mancy et al., 2022*).

We refined the tree-building algorithm to generate trees consistent with the phylogeny. This required creating a pairwise patristic distance matrix from the maximum likelihood phylogeny in R using the ape package (*Paradis and Schliep, 2019*), from which genetic clusters were assigned using the adegenet package (*Jombart, 2008*; *Jombart and Ahmed, 2011*), with a cutoff value of 0.002. Following the steps outlined in *Figure 4—figure supplement 1*, we then built a directed graph of the transmission tree and sequentially sampled edges connecting mismatched genetic clusters to rebuild these paths to generate trees consistent with phylogenetic assignments. First, we sampled by frequency, that is, how often edges occur in paths with mismatches, then by the scaled probability of the dispersal distance and serial interval from the assigned progenitor, generally selecting lower probability links to resample. For edges that were broken, we sequentially resampled a progenitor from those that generated trees consistent with the phylogenetic assignments.

To further resolve transmission chains, we applied additional pruning steps to filter out case pairs where the time interval or distance exceeded the 99th percentile of the serial interval and distance kernel distributions (without pruning or integration of phylogenetic information, the tree reconstruction results in a single large chain). The tree reconstruction methods are wrapped into an R package (available at https://github.com/mrajeev08/treerabid and archived on Zenodo DOI: 10.5281/zenodo.5269062; *Rajeev, 2023*). We compared pruned trees (split into transmission chains) to transmission trees reconstructed to be consistent with the phylogeny. For each pruning algorithm, we compared across the consensus trees (i.e. the most frequently assigned progenitors for each case), the Maximum Clade Credibility (MCC) trees (the tree within the bootstrap that had the highest product of progenitor probabilities) and the majority transmission trees (the tree within the bootstrap that had the highest number of consensus progenitors), shown in *Figure 4—figure supplements 3–5*, respectively.

The effective reproduction number $R_e$, which describes transmission in the presence of control measures, was estimated from the number of secondary cases per case in the transmission trees. We examined $R_e$ over time by fitting a LOESS regression with date of case as our predictor and $R_e$ as our response. We also looked at individual $R_e$ estimates in relation to vaccination coverage at the time of symptoms in the *shehia* where each case occurred (*Figure 2—figure supplement 1*) and compared the distributions of $R_e$ from different tree summaries.

We estimated the case detection achieved from our contact tracing using recently developed analytical methods (*Mancy et al., 2022*; *Cori et al., 2018*). Specifically, we used the times between statistically or directly-linked cases from the transmission tree reconstructions and the serial interval distribution for rabies, to fit the simulated distribution of numbers of unobserved intermediates, assuming all infected individuals have the same probability of being detected. To account for the long-tailed distribution of serial intervals, we sorted simulated values for initial intervals to most closely match observed values (i.e. so long incubators are accounted for and not always taken to be cases with multiple generations separating them from their progenitors). This approach with sorting generally performs better than the unsorted approach (*Mancy et al., 2022*) but tends to underestimate detection probabilities by about 10%, in particular for values between 0.3 and 0.75. We examined the fit across a range of detection probabilities for the endemic period (2010–2014), the subsequent outbreak (2016–2018) and overall, applying the method to 100 bootstrapped trees generated by the pruning strategies (with and without genetic information), and to the majority tree and the MCC tree, taking the mean of 10 estimates as the detection probability for each tree.

## Cost-effectiveness analyses

We used the contact tracing data to inform a probabilistic decision tree model to estimate the impacts and cost-effectiveness of interventions on Pemba (Figure 6). We compared a baseline scenario without dog vaccination and with patients charged for PEP (as was initially the case on Pemba), with scenarios of free PEP provisioning but without dog vaccination, and with both free PEP and sustained island-wide dog vaccination carried out annually, that is, a One Health approach, over a ten-year time

horizon. From compiled cost data (*Supplementary file 2*), we estimated the per campaign cost of island-wide dog vaccination and the per patient cost of PEP for use in the model. We estimated the probability of rabies-exposed bite victims starting and completing PEP (defined as at least 3 doses) from 2010 to 2015 (when most bite victims paid for PEP) and 2016–2020 (when most bite victims received free PEP), and the frequency of healthy dog bite victims presenting for PEP. After adjusting for case detection, we sampled the time series of rabid dogs on Pemba, to generate rabies incidence under scenarios with and without dog vaccination. For scenarios with dog vaccination, we assumed the first campaign took place in year one, translating to reduced incidence from year two onwards, as per the contact tracing data, sampled from 2010 to 2015 and from 2016 to 2020 with zero incidence thereafter. Using negative binomial parameters fitted to the offspring distribution of bite victims per rabid dog, adjusted for case detection, we simulated corresponding time series of rabies exposures. We tuned the simulated incidence of healthy bite patients to match the data under these scenarios. Parameter estimates for probabilities of starting and completing PEP and for rabies progression in the absence of PEP (*Hampson et al., 2019*) were used to estimate deaths and deaths averted. We took the perspective of the health provider and report cost-effectiveness per death averted, with costs discounted at 3%. All monetary values presented are in 2023 US dollars.

## Results

Rabies was endemic on Pemba in 2010 at the start of the study. That year we traced 32 human rabies exposures, 33 rabid dogs and three human rabies deaths diagnosed from clinical signs and history of exposure (6.77 exposures and 0.63 deaths/ 100,000 persons and 10.5 cases/ 1000 dogs). Initial dog vaccination implemented as part of a rabies elimination demonstration project in 2011 achieved only low and heterogeneous coverage (13% in 2011, ranging from 7% to 20% across districts), but by 2014 campaigns were island-wide and achieved better coverage (mean 50%, range 46–60%, *Figure 1*). Correspondingly, human rabies exposures and dog rabies cases declined each year to just 2 each in 2014. The effective reproduction number, $R_e$, also declined from around 1.5 in 2010 to <1 in 2014 (*Figure 2*). No human rabies exposures, deaths or animal cases were detected from May 2014 until July 2016.

In August 2016, an influx of bite patients was seen in hospitals on Pemba. By the year end, we had traced 35 human rabies exposures and 27 dog rabies cases. In response to this outbreak, the Ministry of Livestock and Fisheries Development initiated dog vaccination in *shehias* with recorded dog cases, but because the outbreak spread rapidly, islandwide dog vaccination was reinstated. In 2017, we traced three human rabies deaths, 126 rabies exposures and 62 rabid dogs (26.6 exposures and 0.63 deaths/100,000 people, and 19.6 cases/1000 dogs). High vaccination coverage was achieved consecutively over subsequent annual dog vaccination campaigns from 2017 to 2020 (median 61%, range 46–78% in 2019, *Figure 1*). Incidence rapidly declined from the 2017 peak with 19 human rabies exposures and 8 dog rabies cases detected in 2018. At the start of the outbreak $R_e$ was high (>1.5), but subsequently declined to <1, with all transmission interrupted by October 2018 (*Figure 2*). No human rabies exposures, deaths or rabid dogs have been identified since (as of September 2022).

Phylogenetic analyses indicated considerable viral diversity from 2010 to 2014 (*Figure 3*). Pruning by time split the reconstructed tree into two transmission chains, from 2010 to 2014 and 2016 to 2018, respectively. Integrating the phylogenetic data further split the transmission chains into varying sizes associated with each sampled lineage, consequently resolving five distinct transmission chains from 2010 to 2014 (*Figure 4*, *Video 1*). During this period we detected approximately 54% of dog rabies cases circulating on the island (95% credible intervals (95% CI) 46.4–62.0%, 92 of an estimated 171 rabid dogs 95% CI: 148–198 rabid dogs, *Figure 5*). Further pruning by the dispersal kernel distance threshold created an additional 3–4 unsampled transmission chains and 5–6 orphaned cases, likely because links connecting these cases were either missed or human-mediated. At least one divergent lineage (reported as Tz5 in *Brunker et al., 2020*) is known to have circulated during this period. *Figure 4—figure supplements 3–5* show in grey cases that were not grouped by the current phylogenetic assignment.

Using RABV-GLUE to obtain an alignment of all sequences from the Cosmopolitan-AF1b minor clade up to the year 2017 (inclusive) regardless of genome position or length resulted in a dataset of 2557 sequences (partial and whole-genome) from 21 countries (all sub-saharan Africa, aside from one Thailand sequence) spanning the years 1980–2017 (*Figure 3—figure supplement 1*). This provided a

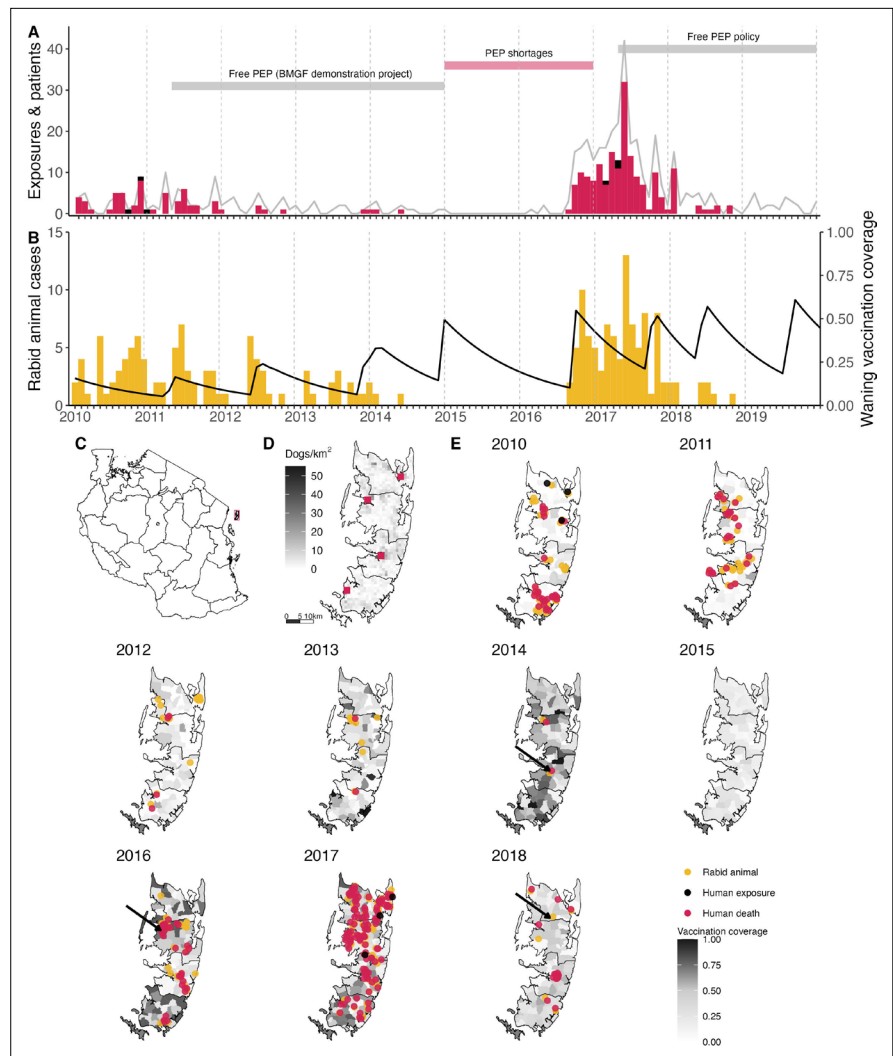

**Figure 1.** Timeline of rabies on Pemba Island in relation to control and prevention measures. (**A**) Monthly time series of traced human rabies exposures (red) and deaths (black), and patients presenting to clinics from bites by both healthy and rabid dogs (grey line). Periods when PEP was provided free of charge are indicated by the grey horizontal bars, as well as periods of shortages (red horizontal bar). (**B**) Dog rabies cases (orange) in relation to average dog vaccination coverage across the island (black line). (**C**) Location of Pemba (red) off the coast of mainland Tanzania. (**D**) Density of Pemba's dog population and location of the four government hospitals that provide PEP (red squares), one in each district. (**E**) Dog rabies cases (orange circles) and human rabies exposures (red circles) and deaths (black circles) each year. Shading indicates dog vaccination coverage in December of each year, projected from the timing of *shehia*-level campaigns, dog turnover and a mean vaccine-induced immunity duration of three years. The arrows point to the last detected animal case in 2014, first detection in the 2016 outbreak and the final case found in 2018.

much wider geographic and temporal context than only the WGS sequences enabling the placement of 2016/17 Pemba outbreak sequences into the context of known background diversity. N and G genes were the most commonly sequenced, with 1042 and 1876 sequences respectively (including WGS), and both gene datasets were used to contextualise Pemba sequences. There was wide variation in the portion of each gene covered and the length of sequences (N: from 203 to full length 1353 basepairs (bp); G: from 277 to full length 1575 bp). The N gene dataset constituted a wider and more even geographic distribution (18 countries), whereas the G gene (16 countries) data was predominantly from South Africa and Tanzania.

Bootstrap values were generally low across the phylogenies, most likely due to the use of short, variable length sequences and therefore should be interpreted with caution. We 'zoomed in' on clusters

within these subtrees to identify the closest relatives to Pemba outbreak cases (*Figure 3—figure supplement 2*). For Pemba cluster 1 (*Figure 3B*, and *Figure 3—figure supplement 2a, b*), sequences were most closely related to sequences from the Serengeti District in northern Tanzania according to both N and G gene datasets. Whereas Pemba cluster 2 sequences (*Figure 3C*, *Figure 3—figure supplement 2c, d*) share a common ancestor with N gene sequences from Zanzibar, a neighbouring island, from the same period (2016/17). This suggests a possible link between rabies outbreaks on these islands and/or a common source of introduction.

Viruses sequenced from the outbreak starting in 2016 belonged to two distinct phylogenetic lineages (*Figure 3*). The time-scaled phylogeny pointed to two independent introductions taking hold and spreading widely, i.e., not continued transmission of viruses circulating previously. Although our estimates of case detection were higher during this outbreak, at 69% (95%CI: 59.4–81.6%, *Figure 5B*),

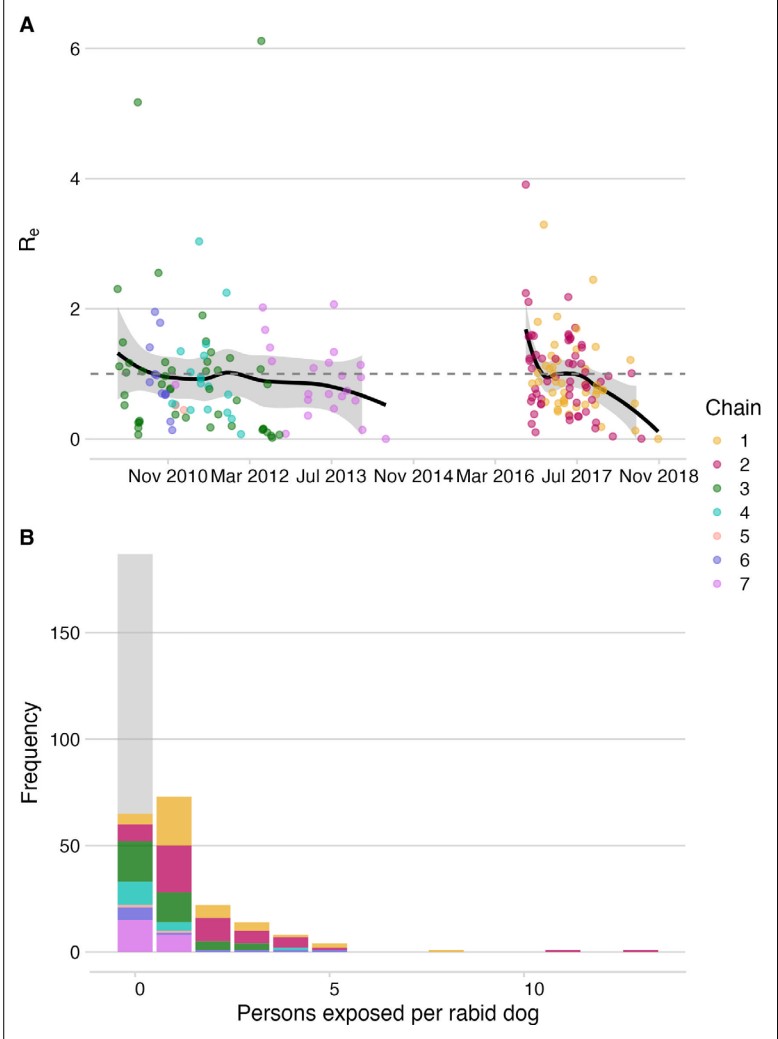

**Figure 2.** Dog-to-dog rabies transmission and dog-to-human rabies exposures on Pemba. (**A**) The effective reproductive number, $R_e$ (black line shows smoothed estimate from a LOESS regression against date of case) with 95% confidence interval (grey envelope) and mean secondary cases from each traced rabid dog inferred from the bootstrapped transmission trees (points). The grey dashed line indicates an $R_e$ equal to 1. (**B**) Inferred offspring distribution of bite victims from rabid dogs. Points/ bars are coloured by transmission chain (see methods and *Figure 4*) with unobserved rabid dogs that did not bite (122 inferred from our estimates of case detection) in grey. A negative binomial distribution fit to the offspring distribution had $\mu$=0.75 and $k$=0.54 (fitting to 2010–2014: $\mu$=0.37, $k$=0.42 and for 2016–2018: $\mu$=1.28, $k$=1.07).

The online version of this article includes the following figure supplement(s) for figure 2:

**Figure supplement 1.** Vaccination coverage versus inferred Re for each case across pruning algorithms.

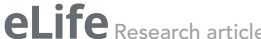

**Figure 3.** Maximum clade credibility tree (MCC) from discrete phylogeographic analysis to identify rabies virus introductions to Pemba. (**A**) Time-calibrated MCC tree of 153 whole-genome sequences from Tanzania, including 13 from the 2016–2018 Pemba outbreak and 6 historical Pemba sequences (2010–2012). Grey vertical bar highlights the window of emergence for the most recent common ancestors of the two introductions that led to the 2016 outbreak (2014.33–2016.29). The expanded subtrees (**B** and **C**) show the Pemba cases one node back from the most recent common ancestor of the 2016 introductions, with branches coloured according to the inferred ancestral location. Black diamonds indicate nodes with >90% posterior support (clade credibilities). Mainland clusters of more than one identical sequence are collapsed. Grey bars represent the 95% highest posterior density interval of node heights, that is estimated age of ancestral nodes. Names of sequences are shown so they can be related to metadata (***Supplementary file 1***).

The online version of this article includes the following figure supplement(s) for figure 3:

**Figure supplement 1.** Rabies virus sequences within the Cosmopolitan-AF1b minor clade.

**Figure supplement 2.** Phylogenetic clusters of RABV sequences from nucleoprotein (N) and glycoprotein (G) datasets showing the closest relatives to Pemba outbreak cases.

i.e. 97 out of an estimated 140 rabid dogs (95%CI: 119–163 cases in outbreak), a few connections linking unsampled transmission chains were still likely missed, with pruning by the dispersal kernel threshold identifying 2–5 unsampled lineages and orphaned cases (***Figure 4—figure supplements 2–5***). Our estimates of $R_e$ over time were consistent irrespective of the transmission tree summary, pruning algorithm or whether reconstructed with or without phylogenetic information (estimates not shown since indistinguishable). Individual variation in rabid dog biting behaviour was identified from contact tracing and inferred for case estimates of $R_e$ (***Figure 2*** also shows variation in persons exposed by individual rabid dogs). On average secondary cases per case declined with higher vaccination coverage in the locality when each case occurred (***Figure 2—figure supplement 1***). While there was uncertainty in exact progenitor assignments, transmission chain assignments were consistent,

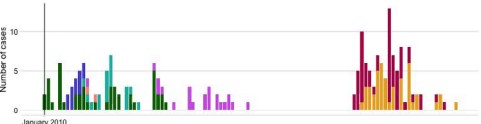

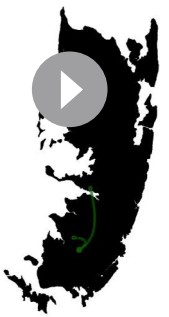

**Video 1.** Rabies cases and inferred transmission chains on Pemba Island. Transmission reconstruction using the consensus links consistent with the phylogenetic assignments. Cases are animated each month, with animals that are incubating infections shown as empty circles until infectious when they transition to filled circles (note that many cases become infectious within the same month of exposure). Inferred transmission links are shown by curved lines and at the approximate time of the exposure event, coloured by transmission chain. Cases identified as introductions are designated by a filled square. The top panel shows the monthly time series of cases by transmission chain.
https://elifesciences.org/articles/85262/figures#video1

and most cases had only a few highly plausible progenitors (>10% chance of being assigned progenitor, *Figure 4—figure supplements 3–6*).

Our approach to case detection inferred from the transmission tree reconstructions were also generally robust, irrespective of the tree summary, pruning algorithm and for the amount of data available. However, our approach tended to underestimate detection probabilities (usually within 10%, *Figure 5A*) at the levels inferred for Pemba (50–70%). That withstanding, we show that given the estimated case detection probabilities, we likely detected most transmission chains comprising more than two cases (*Figure 5C*) and missed only dead-ends or very short chains of transmission.

Of the bite patients presenting to the island's four hospitals from 2010 to 2014 (n=117), a large proportion were bitten by probable rabid dogs (45–72% depending upon the status of unclassified biting dogs), while only a few patients that were bitten by apparently healthy dogs sought care during this period (6.6–12.8 per year, or 1.4–2.7/100,0000 /year). Based on the probability of rabies progression (*Hampson et al., 2019*) and the occurrence of three human rabies deaths in 2010 (*Table 1*), we estimated that 10–31 rabid bite victims did not receive complete or timely PEP and through contact tracing we identified 21 such rabies exposures. The total rabies exposures that we detected from 2010 to 2014 (63-94) were within expectations from triangulating case

detection and rabid dog behaviour (*Hampson et al., 2016*; 65 exposures, range 46–86), and consistent with a 0.66–0.89 probability of rabies-exposed bite victims receiving adequate PEP. During the 2016–2018 outbreak we traced 39 rabid bite victims who did not obtain adequate PEP (late and/or incomplete) and estimated that exposures received appropriate PEP with probability 0.72–0.78 (with the three deaths that occurred early in the outbreak suggesting around 10–31 rabid bite victims did not receive adequate PEP). Probable exposures per rabid dog were higher during the 2016–2018 outbreak than from 2010 to 2014 (1.3 vs 0.34-0.51, both adjusted for case detection) driven in part by variability in dog biting behaviour; two rabid dogs in 2017 each bit more than 10 people (*Figure 2*).

Reasons reported for lack of, or inadequate PEP varied (detailed for deaths in *Table 1*). No PEP shortages were reported whilst PEP was provided for free during the elimination demonstration project (2011–2014). But, at the start of the outbreak in late 2016 patients had to buy PEP (~$12.9 per vial or >$38 for a complete course) and one child bitten in early 2017 by a confirmed rabid dog did not receive PEP due to a shortage. Although too late to be effective, health authorities sought PEP from Zanzibar when the child presented with symptoms, but none was available. In desperation the family took the child to the mainland but with no rabies treatment options they were advised to return home where the child died on arrival. Following the child's death, Zanzibar's Ministry of Health imported PEP and reinstated free-of-charge PEP provisioning. This policy change and sensitization around the outbreak likely contributed to increased health seeking and understanding of the critical need for timely PEP. Contact tracing revealed that 17.5% of rabies exposures were not aware of the importance of PEP early on (2010–2014) compared to <4% during the outbreak (2016–2018) and similarly around 20% of rabies exposures early on (2010–2014), reported not being advised by health workers to obtain PEP, declining to 3% during the outbreak (2016–2018). Patients presenting for healthy dog bites also increased during the outbreak to ~37 /year (7.8/100,000 vs 1.4-2.7/100,000 previously). In



**Figure 4.** Rabies virus transmission chains inferred from epidemiological and phylogenetic data. (**A**) Time series of cases coloured by their transmission chain. (**B**) Consensus transmission tree (the highest probability transmission links that generate a tree consistent with the phylogeny) with chains pruned such that all unsampled cases are assigned to a sequenced chain of transmission. (**C**) Spatial distribution of these cases over the two periods. In (**B**),

*Figure 4 continued on next page*

*Figure 4 continued*

sequenced viruses from sampled cases are indicated by squares with a black outline, while only the tips are shown for unsampled cases. In (**C**), unsampled cases are shown by a filled circle. In all panels, the data are coloured by the transmission chain they were assigned to.

The online version of this article includes the following figure supplement(s) for figure 4:

**Figure supplement 1.** Steps for building transmission trees consistent with phylogenies.

**Figure supplement 2.** Comparing approaches for transmission tree reconstruction.

**Figure supplement 3.** Comparison of the consensus transmission trees (i.e.the most frequently assigned progenitors for each case) across pruning algorithms.

**Figure supplement 4.** Comparison of maximum-clade credibility (MCC) trees (the tree within the bootstrap that had the highest product of progenitor probabilities) across pruning algorithms.

**Figure supplement 5.** Comparison of the majority tree (the tree within the bootstrap that had the highest number of consensus progenitors) across pruning algorithms.

**Figure supplement 6.** Comparison of tree topologies across transmission tree reconstruction algorithms.

the decision tree we used estimates of the probabilities of rabies-exposed bite victims starting and completing PEP for the period 2010–2014 when most patients paid for PEP, and 2016–2020 when most patients received free PEP. Probabilities for starting and completing PEP were 0.667 and 0.397 respectively for 2010–2014 and 0.783 and 0.84 for 2016–2020 respectively (*Figure 6*).

The cost of a complete intramuscular post-exposure vaccination course (4-dose Essen regimen) was approximately $56 versus $25 for an intradermal course (updated Thai Red Cross). Over the 11 years of the study around $17,800 was spent on PEP for 542 bite patients who received a combination of intramuscular and intradermal regimens and had varying levels of compliance. We estimated that this PEP prevented around 42 rabies deaths (95% confidence intervals: 32–55) costing around $424 per death averted. From 2019 onwards, in the aftermath of the 2016–2018 outbreak when all transmission had been interrupted, approximately $876 was spent annually on PEP for patients presenting with bites from healthy dogs (*Figure 1*), i.e., precautionary expenditure post-elimination. Island-wide dog vaccination cost approximately $12,122 per campaign ($13,145 for the campaign that reached most dogs), with a cost of $6.5 per dog vaccinated (range: $4.2–10.8 depending on the campaign). Dog vaccination campaigns interrupted transmission in the dog population within four

**Table 1.** Characteristics of probable human rabies deaths and reported reasons for inadequate Post-Exposure Prophylaxis (PEP).

| Year | Age | Bite site(s) | Type of wound | Reasons for not seeking PEP |
|------|-----|--------------|---------------|------------------------------|
| 2010 | 11–15 years | Both hands and the left palm | Severe wounds with broken bones | After the first hospital visit, the child's family was not advised by health workers to return for subsequent PEP doses and family members were not aware of PEP requirements. |
| 2010 | >50 years | Lower left leg and upper thigh | Deep wounds with multiple tooth penetrations | Victim sought care at a facility (dispensary) that did not provide PEP. Received only first aid without referral to hospital for PEP. |
| 2010 | >50 years | Head (nose) and right arm | Lacerations to nose, large bite to arm, deep tooth penetration | After the wounds healed the victim did not seek their second or subsequent doses of PEP. |
| 2017 | 6–10 years | Neck | Large wound | PEP shortages in Pemba hospitals and prohibitive costs of seeking PEP elsewhere. |
| 2017 | 11–15 years | Face/head and shoulders | Severe wounds that led to hospitalisation | PEP shortages at the hospital where the victim was admitted. Health workers did not advise immediate PEP be sought from elsewhere. |
| 2017 | >50 years | Shoulders, legs, and chest | Large wounds with deep tooth penetrations | Victim thought a single dose of PEP was sufficient for protection and ignored health worker advice to seek subsequent doses. |

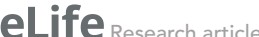

**Figure 5.** Estimation of detection probabilities. (**A**) Estimated detection probabilities from simulated times between linked cases given a known detection probability (x-axis). Colours indicate the number of detected cases used in the simulations. The points show the mean and the lines the range of 10 estimates per simulation. The black dashed line shows the 1:1 line and the grey dashed line the 1.1:1 line. Estimates of detection from these simulations are generally recoverable, although with smaller sample sizes, the estimates are more dispersed. (**B**) Detection probabilities estimated from times between linked cases using the tree algorithm with pruning by the phylogenetic data only. For the estimation, the times between linked cases for a subsample of bootstrapped trees (N=100), as well as the MCC and the majority tree were used. The colours indicate the period for which estimates were generated, 2010–2014 (pre-elimination) and 2016–2018 (reemergence) and overall combining cases. (**C**) Probability of detecting at least one case given estimated detection probabilities and chain sizes (x-axis) with colours corresponding to the period for which estimates were generated.

The online version of this article includes the following figure supplement(s) for figure 5:

*Figure 5 continued on next page*

*Figure 5 continued*

**Figure supplement 1.** Comparison of detection estimates across pruning algorithms and with the inclusion of phylogenetic information.

years of implementation, first in 2014 and again in 2018. However, the lapse in dog vaccination from 2014 allowed the two introductions in 2016 to spread widely.

We parameterized a probabilistic decision tree model and projected rabies incidence, exposures, and deaths under counterfactual scenarios (*Figures 6 and 7*). We estimated that without dog vaccination and with PEP charged to patients (i.e. the status quo prior to the rabies elimination demonstration project) around 27 deaths (95% prediction intervals (95% PIs): 16–39) would occur on Pemba over a 10-year time horizon. On average 48 deaths (95% PIs: 31–67) would be prevented by PEP, at a cost of $300 per death averted (95% PIs: $263–374, with costs discounted at 3%) incremental to a counterfactual without PEP, i.e., in the absence of interventions, 75 human rabies deaths would be expected to occur over ten years on Pemba. Providing PEP for free to patients (as during the rabies elimination demonstration project and by Pemba's government from 2017 onwards) was projected to prevent an additional 10 deaths at a cost of $256 per death averted (95% PIs: $217–333), but still result in 17 rabies deaths (95% PIs: 9–26) over the ten years, with intradermal regimens always more cost-effective than intramuscular regimens. Introducing and sustaining mass dog vaccination, whilst charging for PEP, was projected to prevent 20 deaths relative to the status quo (68 deaths averted overall, 95% PIs: 45–92) costing $1,684 per death averted (95% PIs: $1,264–2,515). Dog vaccination together with free PEP was projected to result in fewest deaths (4 overall, 95% PIs: 1–9), with no deaths after year four (*Figure 7*), and preventing 71 deaths overall (95% PIs: 46–97) at a cost of $1,657 per death averted (95% PIs: $1,228–2,526). Since dog vaccination interrupts transmission, we project that routine dog vaccination would mitigate ongoing risks from introductions and keep Pemba rabies-free, and thus prevent over 300 rabies exposures over the ten years (95% PIs: 263–401) sparing around 30 families

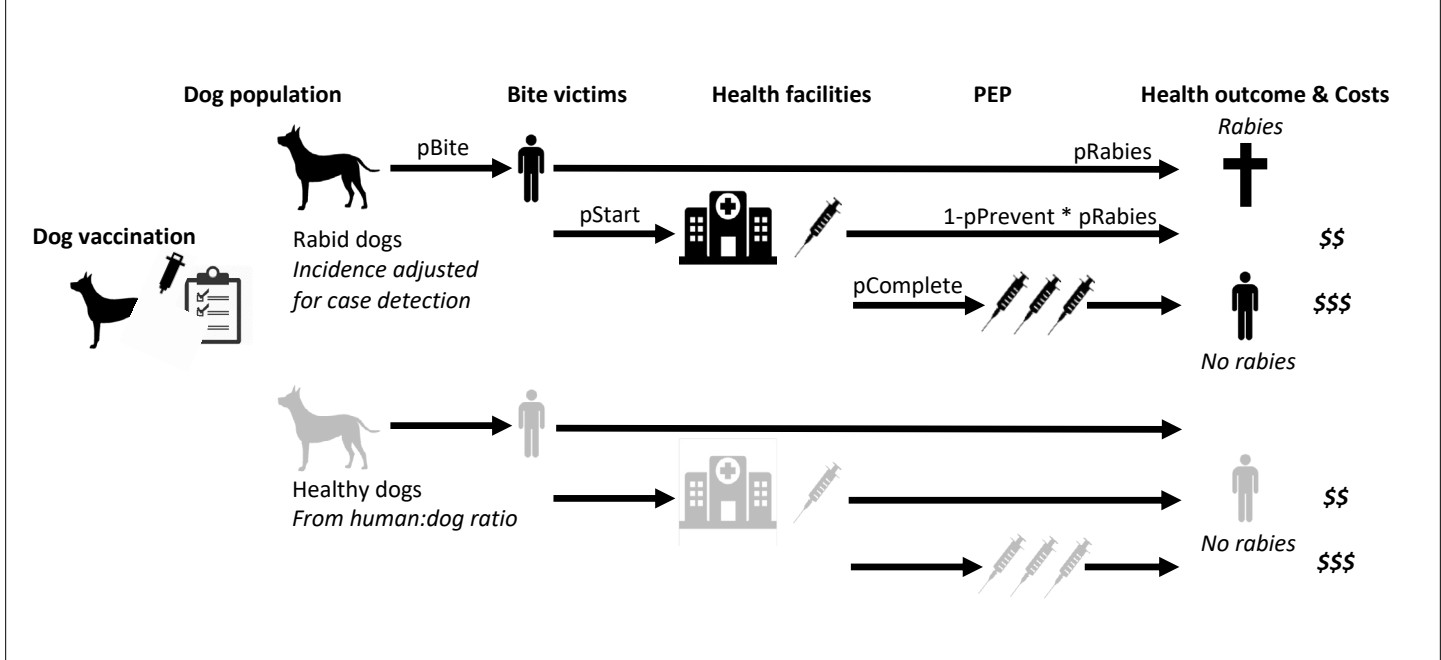

**Figure 6.** Probabilistic decision tree model highlighting mechanisms underpinning health and economic outcomes. Bites per rabid dog (pBite) were drawn from a negative binomial distribution (μ=0.75, k=0.54, fitted to data in *Figure 2B*), while health seeking behaviour of bite victims was modelled to depend on PEP policies. Under free PEP the probability of rabid bite victims presenting and starting PEP (pStart) was 0.783, reducing to 0.667 when PEP was charged for. Healthy bite patients were approximated by 1% of dogs biting per year with the same distribution of bites per dog (pBite) as for rabid dogs when PEP was free, but reduced fourfold (to 0.25%) when patients were charged for PEP. Rabid bite victims developed rabies with probability 0.165 (pRabies) in the absence of PEP and complete PEP was considered 100% effective in preventing rabies, whereas incomplete PEP prevented rabies (pPrevent) with probability 0.986 (*Changalucha et al., 2019*). Dog vaccination determined the trajectory of rabies incidence drawing from case-detection adjusted time series (see modelled time series of rabies exposures, *Figure 7*).

each year from rabid dog bites and the anxiety of needing to urgently obtain life-saving PEP. These results remained robust with and without discounting costs (results not shown).

## Discussion

Dog-mediated rabies is the quintessential zoonotic disease requiring coordinated public health and veterinary interventions as part of a One Health approach to end unnecessary suffering and deaths. Inequities in access to both human and animal vaccines manifest in the continued high burden of rabies in neglected communities around the world. Our study quantifies dog-mediated rabies transmission in an African setting, illustrating how rabies incidence in domestic dogs translates to human rabies exposures and how limitations in provisioning post-exposure vaccines results in human deaths. On Pemba, endemic rabies led to many exposures and three deaths in 2010, the year that our study began. Over the following four years consecutive islandwide dog vaccination campaigns were undertaken during a rabies elimination demonstration project. Initially, only low vaccination coverage was achieved (*Figure 1*), but prevention efforts improved after implementation challenges were overcome, including lack of dog vaccination experience and poor health seeking, conflated by expensive and inaccessible PEP. By 2014, transmission was interrupted. Unfortunately, two independent introductions to Pemba in 2016, at a time when vaccination coverage in the dog population was low, seeded a large outbreak causing three further rabies deaths in 2017. Attempts to respond locally were ineffective, until Pemba's government re-established dog vaccination islandwide, after which rabies was rapidly eliminated. The island has remained rabies-free since October 2018.

Accumulating evidence illustrates how metapopulation dynamics maintain dog-mediated rabies via endemically co-circulating viral lineages (*Mancy et al., 2022*; *Bourhy et al., 2016*). Despite Pemba being a relatively isolated island with a small dog population, genomic data revealed considerable RABV diversity, likely arising from historical introductions, (*Brunker et al., 2015*). The rapid outbreak spread from two contemporary introductions in 2016 highlight the fragility of elimination. While dog rabies remains uncontrolled in nearby populations, reintroduction risks are high (*Bourhy et al., 2016*; *Zinsstag et al., 2017*; *Townsend et al., 2013b*; *Tohma et al., 2016*; *Sharma et al., 2010*). Re-emergence is most likely if dog vaccination coverage is low, causing major public health and economic consequences (*Zinsstag et al., 2017*; *Tohma et al., 2016*; *Windiyaningsih et al., 2004*; *Castillo-Neyra et al., 2017*). Introductions may be reduced through improved border control, but informal human-mediated movement of dogs is not easy to regulate. Scaling up coordinated dog vaccination should suppress the source of introductions and accelerate elimination, accruing and sustaining long-term benefits across much larger populations. While our study from this small island dog population represents a best-case scenario, examples from Latin America show dramatic contractions of dog-mediated rabies when dog vaccination is scaled up and sustained. The last dog-mediated rabies foci on the continent remain only in very poor communities where dog vaccination has been inadequate (*Vigilato et al., 2013*; *Rysava et al., 2020*).

Our detailed contact tracing data from Pemba contrasts with very weak routine rabies surveillance in both humans and animals throughout much of Africa (*Nel, 2013*). Low case detection leads to underestimation of disease burden, lack of prioritisation, and difficulty ascertaining impacts of control, including whether disease has been eliminated, or is circulating undetected. The high case detection on Pemba generated confidence that elimination was achieved, twice (*Townsend et al., 2013a*). The subsequent re-emergence emphasises the need to maintain both surveillance and vaccination coverage where the risk of introductions from connected populations remains. The whole-genome sequences generated in this study further revealed the underlying metapopulation dynamics of rabies circulation. By identifying introductions to Pemba and resolving their role in further spread it was possible to rule out sustained undetected transmission as the cause of re-emergence.

Although contact tracing was intensive compared to routine rabies surveillance which is passive and ad hoc in most African countries, much more intensive approaches are used in high-income countries in response to rabies incursions (*Collective french multidisciplinary investigation team, 2008*), highlighting how dog-mediated rabies is considered a public health emergency. Only one person (KL) completed all contact tracing on Pemba, while also covering other mainland sites (*Lushasi et al., 2021*), in coordination with local livestock field officers, illustrating the capacity of a dedicated epidemiologist/ One Health specialist focusing on rabies. Contact tracing was tractable on Pemba given the relatively low incidence of bite patients (in contrast to some Asian settings, *Hampson et al., 2019*),

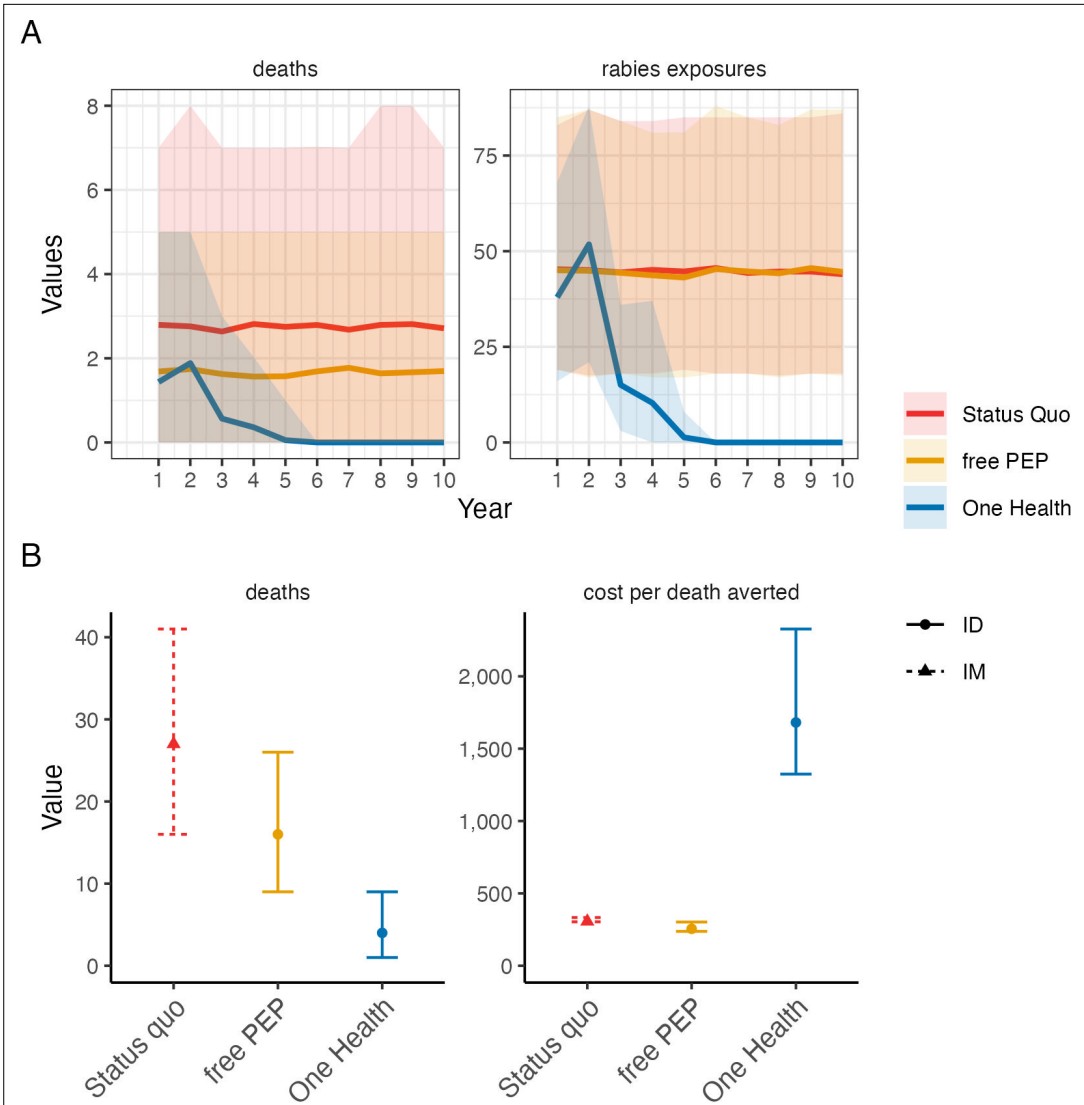

**Figure 7.** Comparison of cost-effectiveness of rabies control and prevention scenarios. (**A**) Projected human rabies deaths (left) and rabies exposures (right) over ten-year time horizon under (i) status quo without dog vaccination and with PEP charged to patient; (ii) free intradermal (ID) post-exposure vaccines, and (iii) a One Health approach with free PEP and routine dog vaccination. Solid lines indicate mean values and shaded envelopes show 95% prediction intervals (PIs). (**B**) Resulting deaths and cost per death averted with 95% PIs. Costs were modelled from estimates of annual island-wide dog vaccination campaigns and of intramuscular (IM) PEP regimens (4-dose Essen, used under status quo) and ID PEP (updated Thai Red Cross, introduced with rabies demonstration project) using data compiled in *Supplementary file 2*.

of which a high proportion were rabies exposures identifiable upon presentation to health facilities. As well as strengthening One Health capacity and intersectoral relationships, contact tracing is a vital tool for investigating emerging infectious diseases, such as Ebola (*Faye et al., 2015*). Contact tracing could be a useful component of rabies control programmes when approaching elimination as caseloads become manageable. Without enhanced surveillance during the endgame, incursions would be more likely to go undetected until human deaths occur (*Townsend et al., 2013b*). Other less intensive approaches, such as integrated bite case management, may be more feasible if used to first increase case detection, strengthen intersectoral working and build technical competence in One Health working generally (*Swedberg et al., 2022*). Genomic approaches are also increasingly affordable, and capacity for genomic surveillance is growing, accelerated over the course of the SARS-CoV-2 pandemic. For rabies, genomic approaches have potential to enhance the information that can

be gleaned from routine surveillance and inform elimination programmes, which are likely to experience such introductions as they progress.

There were limitations in our reconstructions of transmission events despite high case detection. As evident from the tree summaries (*Figure 4—figure supplements 2–6*), we were unable to identify exactly who infected whom within transmission chains, given the low mutation rate of rabies virus and the short intervals between cases. However, we have confidence in the general transmission tree topology and in plausible progenitors within transmission chains (most cases had only a few candidates). Moreover, relevant epidemiological inference, such as estimates of $R_e$ and of outbreak sizes were robust even with uncertainties in transmission pathways and under alternative pruning algorithms. Thus, while improvements in methods to reconstruct transmission trees from both epidemiological and genetic data are warranted, particularly for further characterising spatiotemporal variation in case detection (we assumed all cases had equal probability of being detected), our methods proved valuable. We caveat that while we retrospectively identified a high proportion of cases from their cases histories, we were less successful in recovering samples (~10% of identified cases overall, increasing to >12% during the outbreak). Nonetheless, the genomic data provided key insights that would not have been evident with only epidemiological data: first that rabies was not circulating undetected from 2014–2016 (rabies only re-emerged because of external introductions); second that two introductions occurred in 2016 rather than the outbreak arising from a single introduction (highlighting the non-negligible risk from circulation elsewhere); and finally that it is highly unlikely that other introductions after 2016 established onward transmission (whereas the diversity of lineages circulating pre-2014 indicated multiple introductions previously, *Brunker et al., 2015*). Improved sample recovery and sequencing will be valuable for refining methods, but the relatively low mutation rate of rabies may ultimately limit fine-scale inference, on for example, who infected whom.

Our analyses highlight the cost-effectiveness of PEP as an emergency medicine critical for rabies prevention. We estimated a very low cost per death averted for free PEP provisioning on Pemba (*Figure 7*), even when considered incrementally to the status quo where PEP is charged to patients. Our estimate from Pemba is amongst the highest cost-effectiveness estimates of PEP from across Gavi-eligible countries (*Hampson et al., 2019*) (translating to a cost of just $13 per DALY averted) and results from the high proportion of bite patients presenting with rabies exposures rather than bites from healthy dogs. In settings with more patients seeking care for healthy dog bites, PEP cost-effectiveness declines, although this can be slightly offset by increased vial sharing opportunities under intradermal dose-sparing regimens. Even though PEP is an essential emergency medicine, PEP does not address the suffering caused from injuries inflicted by rabid animals and is insufficient to protect the entire at-risk population. Our study shows how, in practice, lack of awareness, expense and supply issues still prevent access to these emergency vaccines for marginalised populations.

In contrast to PEP sustained mass dog vaccination reduces the risk of exposure and by interrupting transmission in the reservoir can achieve the equitable goal of elimination. Mass dog vaccination inevitably comes at a higher cost per death averted (*Figure 6*) particularly given the relatively high cost per dog vaccinated in this setting. Nonetheless compared to other health interventions (*Bertram et al., 2021*), this One Health approach remains extremely cost-effective. In denser, more connected populations than Pemba rabies elimination is likely to take longer and be more fragile, while conversely the cost per dog vaccinated is likely to reduce in areas with larger dog populations and with opportunities for optimising the delivery of dog vaccination. Our cost-effectiveness estimates lie within expectations for countries in sub-Saharan Africa (*Hampson et al., 2019*), but these considerations limit their transferability. To improve health economic models, the relationship between dog vaccination coverage and risk reduction needs to be better quantified, and research is needed on health seeking behaviours following bites by both healthy and rabid dogs that impact cost-effectiveness. Moreover, realised cost-effectiveness depends on the stochastic nature of outbreaks and the degree to which interventions are delivered as intended, including how dog vaccination coverage is maintained, since vaccination campaigns can often lapse, as seen from Pemba. The COVID-19 pandemic highlights how such disruption can severely set back rabies programmes (*Kunkel et al., 2021*; *Raynor et al., 2021*, *Nadal et al., 2022*).

We conclude that the investment needed for a One Health approach, to support access to life-saving emergency vaccines for bite victims and to achieve and maintain rabies freedom in source populations through dog vaccination is very cost-effective and can bring rapid success. Lessons from

Pemba should build confidence in the feasibility of eliminating rabies elsewhere on the African continent but highlight the importance of sustaining commitment. Coordinated dog vaccination over sufficiently large scales will have the greatest and most long-lasting impacts in equitably tackling this preventable disease.

## Acknowledgements

We are grateful to staff from the Animal and Health Departments on Pemba, Francois-Xavier Meslin from WHO HQ, Pelagia Muchuruza and Alphoncina Nanai from the WHO country office, and local community members for support. Aaron E Lingo, Abdallah N Mauly, Shamata S Khamis, Mcha H Mcha, Abdallah A Mohamed, Hemed M Ali and Abdallah M Sudi all assisted with data collection and Daisy Jennings assisted with sequencing. The Tanzania Ministries of Health and Social Welfare, Livestock Development and Fisheries, the WHO Country Office-Tanzania, the National Institute for Medical Research, the Zanzibar Ministry of Health and Research Council, the Tanzania Veterinary Laboratory Agency, Afrique One-ASPIRE all provided permissions and collaborative expertise. Dr Eberhard Mbunda (deceased) was a champion of rabies control in Tanzania. Professor Rudovick Kazwala (deceased) worked tirelessly for the veterinary and academic sector in Tanzania and was a long-standing inspiring mentor to our team. We greatly appreciate their contributions to this work.

## Additional information

### Funding

| Funder | Grant reference number | Author |
|---|---|---|
| Wellcome Trust | 207569/Z/17/Z | Katie Hampson |
| Wellcome Trust | 103270/Z/13/Z | Kennedy Lushasi |
| UBS Optimus Foundation | | Tiziana Lembo |
| National Institutes of Health (Department of Health and Human Services | R01AI141712 | Katie Hampson |
| DELTAS Africa Initiative | Afrique One-ASPIRE/DEL-15-008 | Kennedy Lushasi |
| Bill and Melinda Gates Foundation | OPP49679 | Sarah Cleaveland |
| UK Department for Environment, Food and Rural Affairs (Defra), Scottish government and Welsh government | SEV3500 & SE0421 | Anthony R Fooks |
| Wellcome Trust | 095787/Z/11/Z | Katie Hampson |

The funders had no role in study design, data collection and interpretation, or the decision to submit the work for publication. For the purpose of Open Access, the authors have applied a CC BY public copyright license to any Author Accepted Manuscript version arising from this submission.

### Author contributions

Kennedy Lushasi, Conceptualization, Data curation, Formal analysis, Investigation, Methodology, Writing - original draft, Project administration; Kirstyn Brunker, Conceptualization, Data curation, Formal analysis, Supervision, Investigation, Visualization, Methodology, Writing - original draft, Project administration; Malavika Rajeev, Formal analysis, Investigation, Visualization, Methodology; Elaine A Ferguson, Rachel Steenson, Formal analysis, Visualization; Gurdeep Jaswant, Joel Changalucha, Anna Czupryna, Msanif Masoud, Ally Z Mohamed, Kassim Omar, Maganga Sambo, Lwitiko Sikana, Investigation; Laurie Louise Baker, Visualization; Roman Biek, Kija Ng'habi, Supervision; Sarah Cleaveland, Funding acquisition, Writing - review and editing; Anthony R Fooks, Resources, Funding

acquisition; Nicodemus J Govella, Supervision, Project administration, Writing - review and editing; Daniel T Haydon, Supervision, Writing - review and editing; Paul CD Johnson, Formal analysis, Supervision; Rudovick Kazwala, Supervision, Funding acquisition, Project administration; Tiziana Lembo, Funding acquisition, Investigation, Writing - review and editing; Denise Marston, Resources, Funding acquisition, Investigation, Writing - review and editing; Matthew Maziku, Eberhard Mbunda, Geofrey Mchau, Investigation, Project administration; Emmanuel Mpolya, Supervision, Investigation; Chanasa Ngeleja, Hezron Nonga, Resources, Investigation; Kristyna Rysava, Investigation, Visualization; Katie Hampson, Conceptualization, Resources, Data curation, Formal analysis, Supervision, Funding acquisition, Investigation, Methodology, Writing - original draft, Project administration

### Author ORCIDs
Kennedy Lushasi ⓘ http://orcid.org/0000-0002-2060-4202
Elaine A Ferguson ⓘ http://orcid.org/0000-0003-2010-765X
Roman Biek ⓘ http://orcid.org/0000-0003-3471-5357
Paul CD Johnson ⓘ http://orcid.org/0000-0001-6663-7520
Emmanuel Mpolya ⓘ http://orcid.org/0000-0002-6210-9445
Katie Hampson ⓘ http://orcid.org/0000-0001-5392-6884

### Ethics

Human subjects: The study was approved by the Zanzibar Medical Research and Ethics Committee (ZAMREC/0001/JULY/014), the Medical Research Coordinating Committee of the National Institute for Medical Research of Tanzania (NIMR/HQ/R.8a/vol.IX/2788), the Ministry of Regional Administration and Local Government (AB.81/288/01), and Ifakara Health Institute Institutional Review Board (IHI/IRB/No:22-2014).

### Decision letter and Author response

Decision letter https://doi.org/10.7554/eLife.85262.sa1
Author response https://doi.org/10.7554/eLife.85262.sa2

---

# Additional files

### Supplementary files

• Supplementary file 1. Epidemiological and next generation sequencing (NGS) metadata. Data detailed for the 22 rabies virus isolates newly sequenced for this study.

• Supplementary file 2. Costs of rabies control and prevention activities. Exchange rate: 1 USD: 2296 Tsh (bank of Tanzania, 05/05/2022 https://www.bot.go.tz/). MoLDF = Ministry of Livestock Development and Fisheries, Tanzania; LTRA = Land transport regulatory authority; DoLD = Department of Livestock Development, Pemba. MSD = Medical Stores Department. LFO = Livestock Field Officer. *We do not include costs of vaccine collection from the airport. **each injection requires 5 minutes of health worker time and up to 8 injections per PEP course.

• MDAR checklist

### Data availability

Code to reproduce the analyses together with deidentified data are available from our public Github repository https://github.com/boydorr/PembaRabies and archived on Zenodo DOI: https://doi.org/10.5281/zenodo.7922464. Sequences are deposited on Genbank.

The following dataset was generated:

| Author(s) | Year | Dataset title | Dataset URL | Database and Identifier |
|---|---|---|---|---|
| Lushasi, et al | 2023 | boydorr/PembaRabies: Integrating contact tracing and whole-genome sequencing to track the elimination of dog-mediated rabies: an observational and genomic study | https://doi.org/10.5281/zenodo.7922464 | Zenodo, 10.5281/zenodo.7922464 |

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
