## [Editor Report]

In this work, the authors set out to use contact tracing and whole-genome sequencing to track the elimination of dog-mediated rabies in Pemba Island, Tanzania. A major strength is the use of multiple data types in the analysis. The work will likely have an impact on influencing the practical policies that can be implemented to target the elimination of dog-mediated rabies in other regions/contexts.

---

## [Decision Letter]

**Decision letter after peer review:**

Thank you for submitting your article "Integrating contact tracing and whole-genome sequencing to track the elimination of dog-mediated rabies: an observational and genomic study" for consideration by *eLife*. Your article has been reviewed by 3 peer reviewers, including Jennifer Flegg as the Reviewing Editor and Reviewer #1, and the evaluation has been overseen by Miles Davenport as the Senior Editor.

Essential revisions:

The following questions need to be addressed:

1) Can the effective reproduction number be estimated from a combination of genetic and contact tracing data? Or can the author comment on the possibility of integrating both types of data into the estimation process?

2) The health economics analysis is too preliminary to draw any strong conclusions from. I think more could be done to improve the health economics methods used.

3) I think the paper oversells a little from its findings. The paper uses a rudimentary health economics approach to make large claims about the cost-effectiveness of different approaches.

4) There has been an important effort to collect all the data presented in this paper. Some of the data are important to better understand transmission dynamics and may not be necessary to monitor and control rabies; some of the datasets may also be expensive to collect, which may be possible to do with the support of research teams but might be impossible to achieve in other settings and on wider areas. I think it would be interesting if authors could discuss how minimal additional data collection would improve the monitoring and control of rabies in an African setting and why accounting for the constraints on cost. I'd also be interested in extending the discussion about the difference between data collected for research (here) and data collected for surveillance.

*Reviewer #1 (Recommendations for the authors):*

1) Can the effective reproduction number be estimated from a combination of genetic and contact tracing data? Or can the authors comment on the possibility of integrating both types of data into the estimation process?

2) The health economics analysis is too preliminary to draw any strong conclusions from. This needs further comment or, ideally, further work.

3) I think the paper oversells a little from its findings. The paper uses a rudimentary health economics approach to make large claims about the cost-effectiveness of different approaches.

4) There has been an important effort to collect all the data presented in this paper. Some of the data are important to better understand transmission dynamics and may not be necessary to monitor and control rabies; some of the datasets may also be expansive to collect, which may be possible to do with the support of research teams but might be impossible to achieve in other settings and on wider areas. I think it would be interesting if authors could discuss how minimal additional data collection would improve the monitoring and control of rabies in an African setting and why accounting for the constraints on cost. I'd also be interested in extending the discussion about the difference between data collected for research (here) and data collected for surveillance.

5) The phylogenetic analyses were difficult to understand. The authors use a phylogenetic framework to estimate the underlying number of rabid dogs per outbreak (171 in the first outbreak and 140 in the second one), but it was unclear to me where the information was coming from. From the supplementary material, it seems the authors build transmission trees consistent with the phylogenies. However, these are reliant on (a) a serial interval and (b) a dispersal kernel. There is no reference as to what serial interval distribution was used and how it was calculated. Similarly, there is no information on the dispersal kernel, including what data was used to fit it. I suspect that the serial interval for rabies (and probably the dispersal kernel) has a long tail, which would lead to substantial uncertainty in the transmission chains, however, I could not see uncertainty in the outbreak sizes.

6) Relatedly, it seems the transmission chain reconstruction relies on identifying missing cases that link observed cases. However, it was unclear to me whether this approach would detect the presence of many dead ends (e.g., an over-dispersed offspring distribution with many cases that don't result in onward transmission) – I could imagine this would lead to similar phylogenetic trees (and probably dispersal kernels) but with a very different number of cases.

7) In Figure S4 panel B, there seems to be a quite even distribution in the probability of linking a pair consistently between 0 and 1 (although I may be incorrect in my interpretation of this figure). Does this not mean that there is ultimately little information in the data on the transmission chains? As these plots are a little confusing to interpret, it would be good to provide guidance as to how to interpret them. I would also provide a written assessment (in the main text) of the authors' ability to accurately reconstruct transmission trees.

8) Given the central importance of the outbreak sizes to the paper – including providing estimates of the case detection probabilities and onwards estimates of the economic impact of the vaccine policies, it would be good to understand the importance of correctly understanding the serial interval and the dispersal kernel to the eventual estimates of the total number of infections. The authors could run sensitivity analyses where these terms are varied.

9) Relatedly, I was not sure of how even case reporting is over space and over time on Pemba. Similarly, it was not clear how representative the sequenced viruses are in space and time (ie was there extra sequencing in some years/some parts of the island). I also wondered whether the fact that sequences come from contact tracing and are therefore not independent samples was problematic. I would set out any key assumptions (and potential limitations) with the phylogenetic approach.

*Reviewer #3 (Recommendations for the authors):*

Relatedly, it seems the transmission chain reconstruction relies on identifying missing cases that link observed cases. However, it was unclear to me whether this approach would detect the presence of many dead ends (e.g., an over-dispersed offspring distribution with many cases that don't result in onwards transmission) – I could imagine this would lead to similar phylogenetic trees (and probably dispersal kernels) but with a very different number of cases.

In Figure S4 panel B, there seems to be a quite even distribution in the probability of linking a pair consistently between 0 and 1 (although I may be incorrect in my interpretation of this figure). Does this not mean that there is ultimately little information in the data on the transmission chains? As these plots are a little confusing to interpret, it would be good to provide guidance as to how to interpret them. I would also provide a written assessment (in the main text) of the authors' ability to accurately reconstruct transmission trees.

Given the central importance of the outbreak sizes to the paper – including providing estimates of the case detection probabilities and onwards estimates of the economic impact of the vaccine policies, it would be good to understand the importance of correctly understanding the serial interval and the dispersal kernel to the eventual estimates of the total number of infections. The authors could run sensitivity analyses where these terms are varied.

Relatedly, I was not sure of how even case reporting is over space and over time on Pemba. Similarly, it was not clear how representative the sequenced viruses are in space and time (ie was there extra sequencing in some years/some parts of the island). I also wondered whether the fact that sequences come from contact tracing and are therefore not independent samples was problematic. I would set out any key assumptions (and potential limitations) with the phylogenetic approach.

---

## [Author Response]

Essential revisions:The following questions need to be addressed:1) Can the effective reproduction number be estimated from a combination of genetic and contact tracing data? Or can the author comment on the possibility of integrating both types of data into the estimation process?

We previously estimated the effective reproductive number (R_e_) from our transmission tree reconstructions that incorporated both the viral whole genome sequences and contact tracing data. We have now updated Figure 2 with a smoothed LOESS regression of R_e_ over time to better visualise the estimate and uncertainty around it. We also moved the methods and results on R_e_ from the supplement to the main text and added some discussion about how the inclusion of both data sources affects our inference, and how future work could build on these approaches.

Specifically in the revision we now highlight that the genomic data reveals that two introductions occurred in 2016 whereas we would have naively expected the outbreak to be the result of a single introduction; and that no other introductions after 2016 established onward rabies transmission on Pemba. These analyses instil confidence that rabies was eliminated not just once but twice confirming the feasibility of elimination through dog vaccination, and reassuringly show that rabies was not circulating undetected from 2014 to 2016 (ruled out by the genomic data) and only re-emerged because of external introductions.

2) The health economics analysis is too preliminary to draw any strong conclusions from. I think more could be done to improve the health economics methods used.

In our revision we have drawn from previously published work to develop a decision tree model of the impact of rabies control and prevention measures (focusing on PEP and mass dog vaccination). We take parameters estimated, where possible, directly from the Pemba data or from the literature e.g. the probability of progression to rabies in the absence of PEP (which was estimated from mainland Tanzania contact tracing data so is expected to be robust in this setting also). Using this model we now project the cost-effectiveness of the different intervention components as part of counterfactual scenarios over a 10-year time horizon. We highlight that a key limitation of our health economics work is the degree to which the conclusions are transferable to other settings, so we also more strongly caveat our findings in the revised discussion and describe other limitations of this approach.

3) I think the paper oversells a little from its findings. The paper uses a rudimentary health economics approach to make large claims about the cost-effectiveness of different approaches.

We agree that our health economics approach was rather rudimentary (see point 2) and that it would be appropriate to be more cautious in generalising from our findings. In our revision we have undertaken a more robust cost-effectiveness analysis, and toned down our claims accordingly to focus on Pemba. We also highlight important uncertainties that should be considered for future work, that would increase the transferability of the approaches we describe.

4) There has been an important effort to collect all the data presented in this paper. Some of the data are important to better understand transmission dynamics and may not be necessary to monitor and control rabies; some of the datasets may also be expensive to collect, which may be possible to do with the support of research teams but might be impossible to achieve in other settings and on wider areas. I think it would be interesting if authors could discuss how minimal additional data collection would improve the monitoring and control of rabies in an African setting and why accounting for the constraints on cost. I'd also be interested in extending the discussion about the difference between data collected for research (here) and data collected for surveillance.

We really appreciate this comment. Our approach was thorough for purposes of research, and hence our data collection was more in-depth than routine government surveillance in Tanzania, which is extremely resource constrained. But, on reflection the contact tracing, outreach and sampling was much less in-depth than in high-income settings (for example in France: https://www.eurosurveillance.org/content/10.2807/ese.13.11.08066-en and the US https://www.forbes.com/sites/brucelee/2021/06/20/rabid-dog-imported-into-us-at-least-12-people-exposed/) and was one of the easier aspects of the study! Lead author KL carried out contact tracing (part-time) in liaison with four livestock officers all equipped with motorbikes for their routine work. However, investment to enable one highly-skilled person to focus on contact tracing rabies has not been considered a priority by LMIC governments or global health funders.

Similarly, development of sequencing protocols was a significant effort (led by KB), but the sequencing from Pemba was small-scale and going forward can be done routinely and cost-effectively by a skilled local scientist (for example, co-author GJ who has since trained several others). Bigger challenges were securing funds for interventions (dog vaccination and free PEP), for training of government animal and public health staff in these interventions and for Tanzanian scientists to undertake postgraduate degrees that led to this manuscript (KL, GJ, MS, LS)! With all this in mind we have clarified in the discussion (paragraph 4) how much can be achieved routinely and make recommendations as to how routine surveillance could be strengthened most efficiently drawing from our experiences.

Reviewer #1 (Recommendations for the authors):1) Can the effective reproduction number be estimated from a combination of genetic and contact tracing data? Or can the authors comment on the possibility of integrating both types of data into the estimation process?

Yes! We previously estimated Re combining these data sources, but both the methodological details and results were in the supplement. We have now moved these to the main text. We also add some discussion about how the two sources of data are integrated and how this adds value to our inference.

2) The health economics analysis is too preliminary to draw any strong conclusions from. This needs further comment or, ideally, further work.

Agreed, we overstated our conclusions. In our revision we developed a more robust decision tree model so as to compare counterfactual scenarios (no intervention, the status quo where PEP is charged to patients, free PEP, and the combination of dog vaccination and free PEP) over a ten-year time horizon, taking the perspective of the health provider. We emphasise the degree to which these scenarios are idealised in comparison to the reality experienced on Pemba, the limitations of this approach and how future work could improve on this. We also point to the specific implications for Pemba and potential transferability of our conclusions to other settings.

3) I think the paper oversells a little from its findings. The paper uses a rudimentary health economics approach to make large claims about the cost-effectiveness of different approaches.

Agreed – see point 2.

4) There has been an important effort to collect all the data presented in this paper. Some of the data are important to better understand transmission dynamics and may not be necessary to monitor and control rabies; some of the datasets may also be expansive to collect, which may be possible to do with the support of research teams but might be impossible to achieve in other settings and on wider areas. I think it would be interesting if authors could discuss how minimal additional data collection would improve the monitoring and control of rabies in an African setting and why accounting for the constraints on cost. I'd also be interested in extending the discussion about the difference between data collected for research (here) and data collected for surveillance.

Thank you for this feedback. We have extended our discussion on these points.

5) The phylogenetic analyses were difficult to understand. The authors use a phylogenetic framework to estimate the underlying number of rabid dogs per outbreak (171 in the first outbreak and 140 in the second one), but it was unclear to me where the information was coming from. From the supplementary material, it seems the authors build transmission trees consistent with the phylogenies. However, these are reliant on (a) a serial interval and (b) a dispersal kernel. There is no reference as to what serial interval distribution was used and how it was calculated. Similarly, there is no information on the dispersal kernel, including what data was used to fit it. I suspect that the serial interval for rabies (and probably the dispersal kernel) has a long tail, which would lead to substantial uncertainty in the transmission chains, however, I could not see uncertainty in the outbreak sizes.

We apologise for these omissions. We have added detail on parameter estimates for the serial interval and dispersal kernel and confidence intervals around the estimated outbreak sizes. We also report our sensitivity analyses where we examine how robust our transmission chains are, and resulting metrics calculated from them, accounting for the long-tailed distributions of both the dispersal kernel and serial interval.

6) Relatedly, it seems the transmission chain reconstruction relies on identifying missing cases that link observed cases. However, it was unclear to me whether this approach would detect the presence of many dead ends (e.g., an over-dispersed offspring distribution with many cases that don't result in onward transmission) – I could imagine this would lead to similar phylogenetic trees (and probably dispersal kernels) but with a very different number of cases.

The reviewer is correct to point out how an overdispersed offspring distribution will lead to many deadends. This was a concern which we examined in some detail and now report in the main text. We were more likely to miss dead-end cases and short chains of transmission (see Figure 5 and supplement), but not sufficiently so to dramatically affect our outbreak estimates. From simulations we estimate that our method was more likely to have underestimated case detection, but not by more than 10%.

7) In Figure S4 panel B, there seems to be a quite even distribution in the probability of linking a pair consistently between 0 and 1 (although I may be incorrect in my interpretation of this figure). Does this not mean that there is ultimately little information in the data on the transmission chains? As these plots are a little confusing to interpret, it would be good to provide guidance as to how to interpret them. I would also provide a written assessment (in the main text) of the authors' ability to accurately reconstruct transmission trees.

These are great points. A limitation from the transmission reconstructions and integration with genomic data is that we cannot identify exactly who infected whom within transmission chains given the low mutation rate of rabies and short spatiotemporal distances between cases. However, we do have more confidence in the high-level topology of our transmission tree reconstructions and plausible progenitors within transmission chains (as most cases had only a few plausible progenitors) which we have explained now in the main text. Probably more important than exact transmission reconstruction is how the method allows us to estimate what proportion of cases from the outbreak were detected and thus determine bounds on the outbreak size. We have added some guidance on interpreting the transmission trees (Figure 4 supplement legends) and assessment of their accuracy in the main results.

8) Given the central importance of the outbreak sizes to the paper – including providing estimates of the case detection probabilities and onwards estimates of the economic impact of the vaccine policies, it would be good to understand the importance of correctly understanding the serial interval and the dispersal kernel to the eventual estimates of the total number of infections. The authors could run sensitivity analyses where these terms are varied.

This is an important point. In the revision we detail our parameter values (serial interval and dispersal kernel) which came from extensive contracting data elsewhere in Tanzania. We also report how robust our estimates of case detection were and thus the uncertainty in outbreak sizes (see point 7). From running various sensitivity analyses of the transmission tree reconstructions (including pruning at differing cuts offs for the serial interval and dispersal kernel distributions) we found these to have very little influence on overall case detection estimates. We explain this in the revision.

9) Relatedly, I was not sure of how even case reporting is over space and over time on Pemba. Similarly, it was not clear how representative the sequenced viruses are in space and time (ie was there extra sequencing in some years/some parts of the island). I also wondered whether the fact that sequences come from contact tracing and are therefore not independent samples was problematic. I would set out any key assumptions (and potential limitations) with the phylogenetic approach.

There was considerable variance in the degree to which cases were identified through contact tracing (typically from their case histories without recovery of a sample) and sequenced (samples were recovered and sequenced from 10% of identified cases, representing approximately 6% of cases estimated to have occurred). For instance, we inferred lower case detection in the initial endemic period (2010-2014, ~54%) than subsequently during the outbreak (2016-2018, 69%). There was no doubt further fine-scale variation in case detection across the island, which we plan to investigate in our continued work on integrating epidemiological and genomic data. However, we think that the work we present gives confidence as to how robust case detection from contact tracing is, with the majority of transmission chains exceeding 2 cases being detected. In our revision we present the robustness of case detection estimates in the main text, and lay out the key assumptions and potential limitations of the approaches that we used.

We do not think that the sequencing of cases identified through contact tracing present a problem in terms of lack of independence. We either combined the two data sources together to draw inference (transmission trees) or conducted the phylogenetic analyses using only the sequences and associated metadata to specifically address the question of introductions. We show in the sensitivity analyses that transmission trees inferred only from epidemiological data (without sequence data) were generally robust with respect to transmission chains (and to a lesser extent to plausible progenitors), but that pruning trees based on time cutoffs increased the congruence of transmission chains identified as consistent with the phylogenetic data. We have elaborated on these points in the revision.